# Genotoxic Activity of Particulate Matter and In Vivo Tests in Children Exposed to Air Pollution

**DOI:** 10.3390/ijerph18105345

**Published:** 2021-05-17

**Authors:** Claudia Zani, Francesco Donato, Elisabetta Ceretti, Roberta Pedrazzani, Ilaria Zerbini, Umberto Gelatti, Donatella Feretti

**Affiliations:** 1Department of Medical and Surgical Specialties, Radiological Sciences and Public Health, University of Brescia, 11 Viale Europa, 25123 Brescia, Italy; claudia.zani@unibs.it (C.Z.); elisabetta.ceretti1@unibs.it (E.C.); ilaria.zerbini@unibs.it (I.Z.); umberto.gelatti@unibs.it (U.G.); donatella.feretti@unibs.it (D.F.); 2Department of Mechanical and Industrial Engineering, University of Brescia, 38 via Branze, 25123 Brescia, Italy; roberta.pedrazzani@unibs.it

**Keywords:** urban particulate matter, children, early biological effects, mucosa buccal cells, micronuclei test, comet assay, polycyclic aromatic hydrocarbons, metals, in vitro mutagenicity, Ames test

## Abstract

The aim of this paper was to investigate the relationship between micronuclei and DNA damage in children’s buccal mucosa cells and the genotoxicity and mutagenicity of the different sized fractions of particulate matter as well as the concentration of PAHs (polycyclic aromatic hydrocarbons) and metals in particulate matter. Air particulate matter was collected by high volume samplers located near the schools attended by the children on the same days of biological samplings. The mutagenic activity was assessed in different cells in in vitro tests (Ames test on bacteria and comet test on leukocytes). Our study showed weak positive correlations between (a) the mutagenicity of the PM_0.5_ fraction and PAHs and (b) the micronuclei test of children’s buccal cells and PAHs detected in PM_0.5_ and PM_0.5–3_ fractions. A positive correlation was also found between in vitro comet test on leukocytes and PAHs in the PM_3–10_ fraction. No correlation was observed for metal concentrations in each PM fraction.

## 1. Introduction

Many studies have shown that urban fine particulate matter (PM) has mutagenic and genotoxic activities in different short-term tests on bacteria and in vitro and in vivo human cells [1,2,3,4,5,6]. The genotoxic and mutagenic activity of PM varies according to its size and physical and chemical composition that depends mainly on local emissions [3,7,8,9,10].

PM_10_ and PM_2.5_ are two of the most used parameters for air pollution evaluation because these fractions carry a wide range of mutagenic, genotoxic and carcinogenic compounds [8].

In vitro mutagenicity/genotoxicity tests on organic extracts of PM are important tools for evaluating the DNA damage effect of this mixture [11] and it is well-known that urban air PM can induce cancer in animals and has mutagenic effects in bacteria, plant and in vitro mammalian cells [2,4,12,13,14,15,16].

Several studies have also investigated the cytogenetic and genotoxic effects of fine PM on in vivo human cells [15,17,18,19]. Particularly, studies carried out on cells derived from target tissues (e.g., pulmonary cells, mucosa cells) showed a positive association with air pollution [20,21,22,23,24,25].

This study is the conclusion of the RESPIRA project that assessed the correlation between air pollution and early biological effects in exposed children. The chemical composition of PM and its genotoxicity and the MN and comet tests on children’s buccal mucosa cells have been reported in previous publications [16,23,26].

This paper aims to assess the relationships between the concentration and toxic equivalency of polycyclic aromatic hydrocarbons (PAHs) in different sized PM fractions collected near the schools attended by children and air sample genotoxicity and micronuclei and DNA damage (comet test) in children’s buccal mucosa cells.

The main novelty of this research is the use of these measures of the early biological effect in a considerable number of young children living in a heavily polluted area, which can be detected a long time before clinical disease develops. 

This study might add both theoretical and practical information on the effects of various air pollutants on children’s health and provide health policy makers with a valuable tool for estimating the burden of disease attributable to urban air pollution. Due to the very large number of people exposed to air pollutants, even a small increase in the risk of disease is a relevant public health issue.

## 2. Materials and Methods

We carried out a molecular cross-sectional study named the RESPIRA study (Italian acronym for Rischio ESPosizione Inquinamento aRia Atmosferica). We evaluated the DNA damage according to air pollution exposure in a sample of 132 pre-school children aged 3–6 years living in Brescia, a highly polluted town in Northern Italy, using the comet assay for salivary leukocytes and the micronuclei (MN) test for buccal mucosa cells [23,26]. The children’s exposure to urban air pollution was evaluated by collecting PM samples in the six school areas. A high volume air sampler (Air Flow HVS PM_10_, AMS Analitica Srl, Pesaro, Italy) with a 1.13 m^3^/min flow was located near each school for 48 h in order to collect different sized fractions of PM_10_ during the same days of the biological sampling. Ultrafine, fine and coarse particles were sampled on fiberglass filters submitted to a gravimetric determination of PM_0.5_ (PM diameter < 0.5 µm), PM_0.5–3_ (PM diameter between 0.5 and 3 µm) and PM_3–10_ (PM diameter between 3 and 10 µm). The air samples (six total samples of PM) were collected on different working days for each school in winter from January 2012 to March 2013. 

The concentration of total PAHs and metals in the PM was determined in the PM samples collected near the children’s schools on the same days of biological sampling. Sampling and analytical procedures together with the obtained results are detailed in a previous paper where the genotoxic activity of particulate matter and its possible correlations with PAHs and metals were assessed [16].

The organic extract of different fractions of PM was analyzed for: (1) PAH and metal concentrations; (2) in vitro mutagenicity on bacteria with the *Salmonella*/microsome assay (Ames test); (3) in vitro genotoxicity on human leukocytes with the comet assay [16]. 

We computed the increase of mutagenic activity for each m^3^ of air for all of the bacteria strains using the Ames test [16]. In this paper, the same analysis was performed using the comet test on leukocytes expressed as an increase of the visual score for each m^3^ of air. 

Regarding the PAHs, we calculated the concentration of total carcinogenic molecules as the sum of the compounds with definite or possible carcinogenic activity according to the IARC classification: benzo[a]anthracene (BaA), chrysene (CHR), benzo[b]fluoranthene (BbF), benzo[k]fluoranthene (BkF), benzo[j]fluoranthene (BjF), indeno[1,2,3-cd]pyrene (IcdP), dibenzo[a,h]anthracene (DahA), dibenzo[a,l]pyrene (DBalP), dibenzo[a,i]pyrene (DBaiP), dibenzo[a,h]pyrene (DBahP).

Furthermore, we computed the toxic equivalency (TEQ), as usually done for compounds with several congeners (for instance, dioxins), to assess the overall effect of the chemical mixture using the most toxic one as the reference with the toxicity set to 1 based on epidemiological and toxicological literature data.

In particular, we calculated: (1) the total carcinogenic equivalency, CEQ (as proposed by Rogula-Kozłowska et al. [27] after Nisbet and LaGoy [28]), based on in vivo carcinogenicity data; (2) the toxic equivalency concentration, TEQ-DFG, according to the German Research Foundation (DFG) where a similar approach was followed but updated and improved to include more congeners [29]; (3) the toxic equivalency of the mixtures, TCDD-TEQ, referred to as the induction potency exerted by 2,3,7,8-tetrachlorodibenzo-p-dioxin (TCDD) [30]; (4) the BaP potency equivalence, BaP-PEQ, as indicated by the Minnesota Department of Health (MDH) [31]; (5) the mutagenic equivalence, MEQ, calculated as reported by Rogula-Kozłowska et al. [27]. The details of the calculations are reported in the mentioned previous paper where the toxic potentials of each sized fraction were compared [16].

A non-parametric correlation coefficient (Spearman rho) was used to assess the possible correlation between the concentration of total PAHs, PAHs with carcinogenic activity and BaP and metals in in vitro tests for each PM fraction. We also evaluated the relationship between the in vivo comet and MN tests carried out on children’s buccal cells and total PAHs, carcinogenic PAHs and BaP, PAH toxic equivalencies computed and metal concentrations. A linear regression analysis was performed to assess the associations investigated, adjusting for PM fractions and the days of biological sampling. Two-tailed statistical tests were applied, with a 0.05 *p*-value as the threshold for rejecting the null hypothesis. All of the analyses were performed using the Stata TM 12.0 statistical package (Stata Statistical Software Release 12.0, 2012; Stata Corporation, College Station, TX, USA).

## 3. Results

In Table 1, the increase of DNA damage in an in vitro comet test on leukocytes per each m^3^ of air collected according to the site of sampling and the PM fraction is reported. The results of the in vitro comet test on leukocytes showed that the largest increase of mutagenicity for each m^3^ of air was in the PM_0.5_ compared with the other PM fractions.

Table 2 reports the concentration of PAHs with carcinogenic activity according to the PM fraction for each sampling site and all sites together. The PM_0.5_ fraction contained the highest level of PAHs with carcinogenic activity (3.6 ng) compared with the other fractions (0.7 and 2.2 ng in PM_0.5–3_ and PM_3–10_, respectively).

The highest levels of several metals analyzed were found in the PM_3–10_ fraction (Fe, Al, Mn, Zn, Cd and Pb) whereas the concentrations of other metals were particularly high in the PM_0.5_ fraction (Cr, V, Ni, Hg) (Table 3).

In Table 4, the values of PAH toxic equivalency measures for each PM fraction are reported. The smallest PM fraction (PM_0.5_) showed the highest values of all of the PAH toxic equivalency measures.

In Figure 1, Figure 2 and Figure 3, we report the positive linear regressions found between the PAH concentrations (total PAHs, carcinogenic PAHs and BaP) and in vivo and in vitro test results for each PM fraction. No significant correlation with metal concentration was observed (data not reported). In detail, the Ames test on TA98 without metabolic activation showed a weak correlation with concentrations of total PAHs, carcinogenic PAHs and BaP in PM_0.5_ (Figure 1). On the contrary, the in vitro comet test on leukocytes that tested the organic extract of PM showed a fair positive linear relation with total PAHs, PAHs with carcinogenic activity and BaP in the PM_3–10_ fraction only (Figure 2).

The micronuclei test of children’s buccal cells showed a weak positive relation with total PAHs, carcinogenic PAHs, BaP and PAH toxic equivalency in both PM_0.5_ and PM_0.5–3_ fractions. Linear regressions of micronuclei on total PAHs, carcinogenic PAHs and BaP in the PM_0.5_ fraction are reported in Figure 3. The coefficients of the linear regression and their standard errors (SEs) for statistically significant correlations only are reported in Table 5.

No relationship was observed between the in vivo comet test of children’s buccal cells and total PAHs, PAHs with carcinogenic activity and BaP for each PM fraction; neither the in vivo nor the in vitro tests were related with the concentration of metals (data not shown).

As the concentration of total PAHs, PAHs with carcinogenic activity and BaP was higher in air samples taken in site 1, where the school was located near a steel plant, we compared the MN test of the children attending this school with all of the others together. In Figure 4, the box-and-whisker graphs describe the minimum, maximum, median and first and third quartiles of micronuclei distribution in children’s buccal cells. Using the non-parametric Mann–Whitney test we found that the differences in micronuclei frequency between the children attending school 1 and those attending the other schools together were not statistically significant.

## 4. Discussion

The results of the in vitro tests applied in this project (the Ames test on bacteria and the comet test on human leukocytes) were converted into an increase of mutagenicity per each m^3^ of air (corresponding with the dose increase) to evaluate the mutagenic activity in the different fractions of PM and showed that the smallest PM fractions usually had the greatest impact. We also found that only the PM_0.5_ and PM_0.5–3_ fractions showed an increase in mutagenicity per each m^3^ of air for the Ames test [16]. The PM_3–10_ fraction never showed an increase in genotoxic activity, in agreement with previous studies [12,21,32,33]. The comet test on leukocytes (in vitro test) also showed the highest increase of DNA damage per each m^3^ of air in the smallest fraction.

With regard to the in vitro tests, the Ames test on the TA98 strain only (without metabolic activation) was slightly related to the total and carcinogenic PAHs and BaP concentration although a recent study also found a relationship between mutagenicity and PAH levels with other bacterial strains [34].

The comparison between the children attending the school in an area with high levels of air pollutants due to the proximity to a steel plant and those attending other schools showed that the former had higher DNA damage in their buccal cells than the others although the difference was not statistically significant. The PM_0.5_ and PM_0.5–3_ fractions are usually those with the greatest toxicity especially due to organic compounds such as PAHs, as reported by other authors [3,35,36,37]. Indeed, the smallest PM fractions are the most toxic probably because they are richer in PAHs and have a higher equivalent toxicity [16]. In this study we also observed an association between the early biological effects in children and the smallest PM and PAH concentrations. BaP as well as other PAHs can play critical roles in children, inducing mutagenic effects and contributing to physiological alterations and epigenetic alterations [37,38,39].

No metal concentration was related with the in vivo tests on the children’s buccal mucosa cells although for a few elements other studies have shown a relationship [40,41]. Several metals are capable of damaging DNA and can be pro-mutagenic [42,43]; their content in PM fractions appeared not be associated with in vitro genotoxic effects on bacteria and leukocytes in our study. On the other hand, the Ames test does not detect several oxidizing mutagens and cross-linking agents [44]. The PM_3–10_ fraction contained higher concentrations of several metals than the smallest fractions, several of which are harmful to humans (As, Pb, Cr, V, Cd, Ni) but not all of them exhibit mutagenic activity [42]. A few of these, particularly abundant in this fraction such as Cr, V, Cd, Ni and Fe, can induce DNA damage mainly through the formation of reactive radicals and may also inhibit the DNA repair processes. The metal concentration in PM depends on temperature, air circulation and wind speed and may be influenced by pollution levels, modifying the composition of PM [45] and this can affect the response to mutagenicity tests.

The in vivo tests on children showed that DNA damage measured as increase of MN was better associated with air pollutants (PAHs) in different fractions of PM than the DNA damage assessed by the comet assay of children’s mucosa buccal cells. Similar results have been reported by other authors, highlighting that MN and comet tests detect independent effects [46,47]. The MN test was more sensitive for detecting early genotoxic damage in children exposed to air pollutants and PAH exposure was associated with higher MN frequencies in several population studies. In our study, we found that the in vivo tests for detecting an early biological effect in children were associated with the concentration of several air pollutants and therefore they could be used as simple, rapid and low-cost tests for evaluating specific environmental situations and also the impact of possible interventions for contrasting the health effects of air pollution. In particular, the MN test, being predictive of the presence of health effects in children, could be proposed in addition to the traditional parameters for the monitoring of urban air pollution.

## 5. Conclusions

We found a slight positive correlation between total PAHs, PAHs with carcinogenic activity and BaP and mutagenicity in in vitro tests on bacteria in the PM_0.5_ fraction and with the micronuclei test of children’s buccal cells in the PM_0.5_ and PM_0.5–3_ fractions. A positive correlation was found also between early DNA damage in the in vitro comet test and PAHs in the PM_3–10_ fraction. No relationship was observed between metal concentrations and biological in vivo and in vitro tests for all PM fractions.

Together with in vitro genotoxicity tests on PM, the MN test on children’s mucosa buccal cells as a biomarker of early biological effects can be a useful tool for monitoring the risk of genetic damage due to air pollution exposure.

## Figures and Tables

**Figure 1 ijerph-18-05345-f001:**
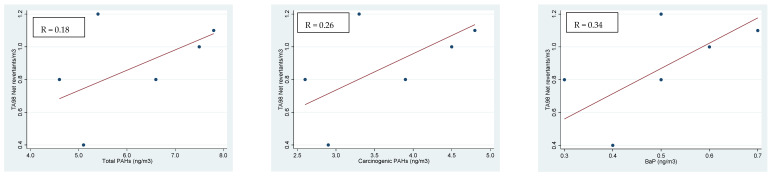
Linear regressions of mutagenicity in the Ames test with the TA98 strain on total PAHs, carcinogenic PAHs and BaP concentration in PM_0.5_ and the Spearman’s correlation coefficients (*p* < 0.001 for all comparisons).

**Figure 2 ijerph-18-05345-f002:**
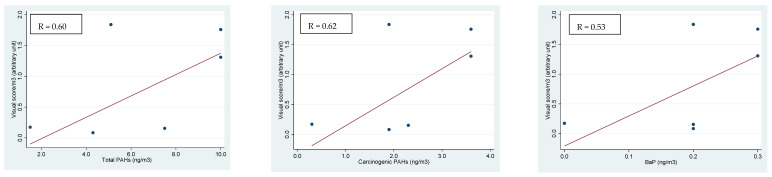
Linear regressions of DNA damage in leukocytes (comet in vitro) on total PAHs, carcinogenic PAHs and BaP concentrations in PM_3–10_ and the Spearman’s correlation coefficients (*p* < 0.001 for all comparisons).

**Figure 3 ijerph-18-05345-f003:**
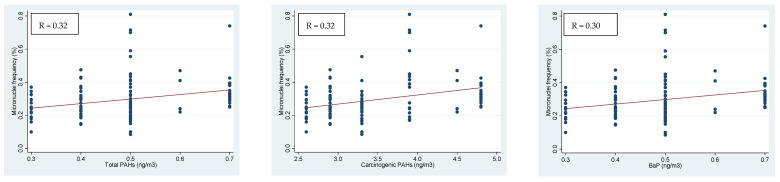
Linear regressions of micronuclei in children’s buccal cells on total PAHs, carcinogenic PAHs and BaP concentrations in PM_0.5_ and the Spearman’s correlation coefficients.

**Figure 4 ijerph-18-05345-f004:**
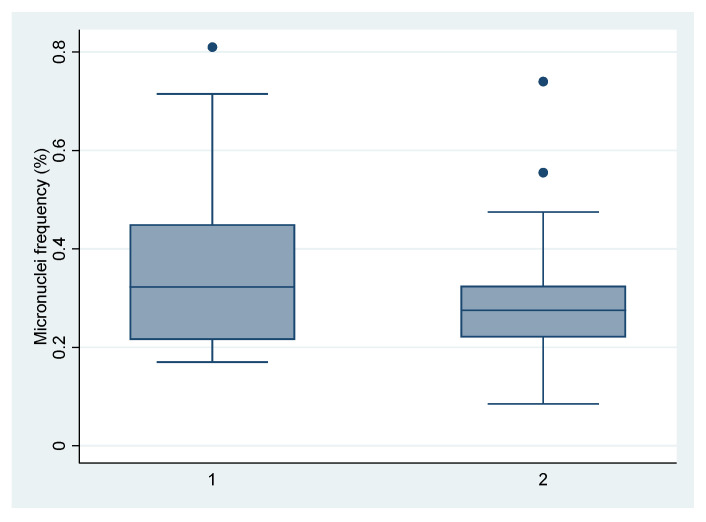
Micronuclei levels in children’s buccal cells in children attending school 1 (near a steel factory) and those attending schools at other sites.

**Table 1 ijerph-18-05345-t001:** Results of an in vitro comet test on leukocytes expressed as an increase of DNA damage per each m^3^ of air according to the sites and PM fractions.

Sites	PM Fractions (µm)	Comet Test on Leukocytes Visual Score
1	<0.5	1.8
	0.5–3	1.6
	3–10	0.1
2	<0.5	2.4
	0.5–3	1.1
	3–10	1.3
3	<0.5	2.7
	0.5–3	1.7
	3–10	1.7
4	<0.5	3.4
	0.5–3	2.2
	3–10	1.8
5	<0.5	3.2
	0.5–3	1.0
	3–10	0.1
6	<0.5	3.1
	0.5–3	1.1
	3–10	0.1

**Table 2 ijerph-18-05345-t002:** Concentrations of PAHs with carcinogenic activity (IARC classification) measured at each sampling site in different fractions of PM.

Sites	PM Fractions (µm)	Carcinogenic PAHs (ng)
1	<0.5	3.9
	0.5–3	3.9
	3–10	2.3
2	<0.5	2.9
	0.5–3	0.1
	3–10	3.6
3	<0.5	3.3
	0.5–3	0.1
	3–10	3.6
4	<0.5	4.5
	0.5–3	0.1
	3–10	1.9
5	<0.5	4.8
	0.5–3	0.1
	3–10	1.9
6	<0.5	2.6
	0.5–3	0.0
	3–10	0.3
Mean ± SD	<0.5	3.6 ± 0.8
	0.5–3	0.7 ± 1.5
	3–10	2.2 ± 1.2

**Table 3 ijerph-18-05345-t003:** Concentrations of metals measured in different fractions of PM.

	PM Fractions (µm)	Metals (ppb)										
		Fe	As	Al	V	Cr	Mn	Ni	Zn	Cd	Hg	Pb
Mean ± SD	<0.5	909.7 ± 1536.9	1.2 ± 0.5	2270.3 ± 1302.6	1.8 ± 1.5	161.9 ± 371.5	35.2 ± 31.4	75.6 ± 171.3	500.4 ± 302.2	0.4 ± 0.2	0.7 ± 0.6	20.8 ± 8.4
	0.5–3	1078.2 ± 259.6	0.4 ± 0.1	5223.4 ± 2580.5	1.2 ± 0.3	18.4 ± 2.2	40.1 ± 9.2	5.9 ± 1.6	17,839.6 ± 14,895.4	4.6 ± 3.7	0.3 ± 0.07	36.7 ± 21.4
	3–10	1256.7 ± 358.1	1.2 ± 0.2	7480.9 ± 3706.5	1.5 ± 0.3	22.0 ± 2.4	63.0 ± 13.6	9.1 ± 0.6	27,150.6 ± 22,141.8	7.2 ± 5.4	0.3 ± 0.08	69.8 ± 32.7

**Table 4 ijerph-18-05345-t004:** Various PAH toxic equivalency measures for each PM fraction according to the site of sampling.

Sites	PM Fractions (µm)	BaP-PEQ ng/m^3^	CEQ ng/m^3^	MEQ ng/m^3^	TCDD-TEQ µg/m^3^	TEQ ng/m^3^
1	<0.5	2.985097	0.49464	1.936413	4.20021	1.258376
	0.5–3	0.223235	0.012657	0.156956	0.13165	0.168213
	3–10	1.207709	0.18187	0.672044	2.33371	0.535057
2	<0.5	2.505132	0.409552	1.624128	3.6108	1.013315
	0.5–3	0.232974	0.012972	0.158597	0.13692	0.171557
	3–10	1.781841	0.277915	0.948737	3.9429	0.692266
3	<0.5	2.860301	0.46415	1.898558	3.93091	1.181743
	0.5–3	0.232694	0.015112	0.165548	0.13395	0.176227
	3–10	1.518079	0.223874	0.965435	3.20357	0.722432
4	<0.5	2.402421	0.453998	1.477876	5.49091	1.057894
	0.5–3	0.202135	0.01118	0.143572	0.08954	0.158247
	3–10	1.006692	0.135052	0.5868	2.11792	0.455188
5	<0.5	2.769771	0.60323	1.690773	5.62284	1.233482
	0.5–3	0.21174	0.011946	0.152389	0.10247	0.165811
	3–10	1.142164	0.214775	0.595705	2.15638	0.47243
6	<0.5	1.602095	0.344192	0.960746	3.27267	0.704601
	0.5–3	0.179467	0.007851	0.136448	0.03739	0.153379
	3–10	0.401406	0.046982	0.242765	0.57702	0.222601

BaP-PEQ = BaP potency equivalence; CEQ = total carcinogenic equivalency; MEQ = mutagenic equivalence; TCDD-TEQ = toxic equivalency referred to in the induction potency exerted by TCDD; TEQ = toxic equivalency concentration.

**Table 5 ijerph-18-05345-t005:** Coefficients of the linear regression (± SE) of MN in children’s mucosa buccal cells according to the concentration and various measures of toxic equivalency of PAHs in each PM fraction.

PM Fractions (µm)	Coefficient ± SE	*p*
<0.5		
Total PAHs (ng)	0.038 ± 0.008	<0.001
Carcinogenic PAHs (ng)	0.054 ± 0.013	<0.001
BaP (ng)	0.274 ± 0.082	0.001
BaP-PEQ (ng/m^3^)	0.06 ± 0.02	0.01
CEQ (ng/m^3^)	0.46 ± 0.12	<0.001
MEQ (ng/m^3^)	ns	
TCDD-TEQ (µg/m^3^)	0.045 ± 0.012	0.001
TEQ (ng/m^3^)	0.17 ± 0.05	0.002
0.5–3		
Total PAHs (ng)	0.016 ± 0.004	0.001
Carcinogenic PAHs (ng)	0.026 ± 0.007	0.001
BaP (ng)	0.19 ± 0.05	0.001
BaP-PEQ (ng/m^3^)	ns	
CEQ (ng/m^3^)	ns	
MEQ (ng/m^3^)	ns	
TCDD-TEQ (µg/m^3^)	ns	
TEQ (ng/m^3^)	ns	

BaP-PEQ = BaP potency equivalence; CEQ = total carcinogenic equivalency; MEQ = mutagenic equivalence; TCDD-TEQ = toxic equivalency referred to in the induction potency exerted by TCDD; TEQ = toxic equivalency concentration; ns = not statistically significant.

## Data Availability

The data presented in this study are available on request from the corresponding author.

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
