# Peer review of "Genotoxic Activity of Particulate Matter and In Vivo Tests in Children Exposed to Air Pollution"

_ijerph, 2021, doi:10.3390/ijerph18105345_

Round 1

Reviewer 1 Report

The research paper presented for review  concerns a genotoxic activity of particulate matter. The work is well constructed (wide description of ‘’Materials and Methods”, “Results”, “Discussion” sections).

However, I haven't noticed in the text the clearly stated novelty of the work (it is very important element of every research) - so some clarification is needed. I think  you should to rebuild the “Introduction” section – to determine the gaps worth studying.

Why did you decide to use the linear regression in your analysis? What values of regression coefficients were obtained? What is their accuracy (standard deviation) and interpretation in context of presented research? It is also worth to pay more attention to the results in the graphical form  (Figures 1-4) - each axes should be labeled to explain what it represents.

The presented work is interesting and includes a valuable study workshop. The amount of research done is impressive.

Author Response

Responses to reviewer

Thanks to the reviewers for the useful comments that allow us to improve the quality of our paper by adding new data.

The reviewers’ comments are in italics. Sentences that are now included in the revised version of the paper are in inverted commas and highlighted in yellow in the manuscript. We have followed almost all the reviewers’ suggestions and modified the paper and references accordingly. 

Comments and Suggestions for Authors

The research paper presented for review concerns a genotoxic activity of particulate matter. The work is well constructed (wide description of ‘’Materials and Methods”, “Results”, “Discussion” sections).

However, I haven't noticed in the text the clearly stated novelty of the work (it is very important element of every research) - so some clarification is needed. I think you should to rebuild the “Introduction” section – to determine the gaps worth studying.

We modified the lines 42-44 of the old version of Introduction section adding the following sentences:

“This study is the conclusion of the RESPIRA project that assessed the correlation between air pollution and early biological effects in exposed children. The chemical composition of PM and its genotoxicity, and the MN and comet tests on children’s buccal mucosa cells were reported in previous publications [16, 23, 26].

This paper aims to assess the relationships between the concentration and toxic equivalency of polycyclic aromatic hydrocarbons (PAHs), in different sized PM fractions collected near the schools attended by children, and air sample genotoxicity, and micronuclei and DNA damage (Comet test) in children’s buccal mucosa cells.

This study might add both theoretical and practical information on the effects of various air pollutants on children’s health and provide health-policies makers with a valuable tool for estimating the burden of disease attributable to urban air pollution. Because of the very large number of people exposed to air pollutants, even a small increase in the risk of disease is a relevant public health issue.”

Why did you decide to use the linear regression in your analysis? What values of regression coefficients were obtained? What is their accuracy (standard deviation) and interpretation in context of presented research? It is also worth to pay more attention to the results in the graphical form (Figures 1-4) - each axes should be labeled to explain what it represents.

We used the linear regression in order to assess the dose-effect relationship between the concentration of PAHs and DNA damage in buccal cells. However, since there were only six sampling sites (school areas) and the children were grouped on the basis of the attended school, we used non-parametric correlation coefficient (Spearman rho), as reported in Materials and Methods.

Lines 74-75 of the old version have been modified as follows: “Non-parametric correlation coefficient (Spearman rho) was used to assess possible correlation ….”.

Due to the non-normal distribution, we used the non-parametric Mann-Whitney test to evaluate the difference in micronuclei frequency between the children attending one school located next to a steel plant and those attending the other schools all together, due to consistently worse air quality parameters in the former than latter areas (Figure 4).

Lines 147-149 of the old version have been modified as follows: “Using the non-parametric Mann-Whitney test we found that the differences in micronuclei frequency between the children attending school 1 and those attending the other schools together were not statistically significant.”

The Figure 4 was simplified to focus on the differences in micronuclei frequency between the two groups of children.

We modified Figures 1-3 adding the axis labels and measure units as suggested by reviewer.

The correlation coefficient R (Spearman rho) and p values have been added to the Figures 1-3. The coefficients of linear regression ± standard error (SE) and p values were reported for each linear regression computed of MN in children mucosa buccal cells according to PAHs in the revised version in Table 5.

The presented work is interesting and includes a valuable study workshop. The amount of research done is impressive.

We thank the reviewer for his comment.

Reviewer 2 Report

The paper aims to address a significant issue, that is the carcinogenicity of childhood exposures to ambient particle-bound PAHs and heavy metals. It appears to be based on a human-subjects research (although not clear) that involves collection of biospecimens in the upper respiratory track from school-age children and measurements of size-fractionated particulate PAHs and heavy metals in the community. Unfortunately, the data are poorly analyzed. In addition, the study and data summaries are also poorly presented, diminishing the scientific value of the study. Specific examples include

1 The lack of description of the study design. A paper is cited [i.e. 26] however, a greater level of detail must be given for the readers to understand the study design and interpret the findings. It is not clear how samples (air and biospecimens) are collected, when, where, how many (schools and children in each school) and additional relevant information (such as co-exposures, diet, secondhadn exposure)

2. The relevance of upper respiratory cells is not explained and the potential for contributions from other exposure pathways such as ingestion. It is known that due to their small size, particulate PAHs and heavy metals are less likely to be deposited in the upper respiratory track. On the other hand, cells in the oral region are regularly exposed to chemicals through digestion.

3. The description of the methods for the toxicological tests (replicates, sensitivity) and chemical analysis is missing. There are no information how PAHs were measured or heavy metals. This is particularly concerning, iven that PAHs concentrations are given in increments of 0.5 ng/m3 is an incosistent manner between Tables (table 2) and Figures.

4. There are no indicators of repeated measurements (i.e. standard error) for both biological measurements and chemicals analysis. Is it because one sample was collected, or analyzed, no replicates.

Overall, the poor presentation and description of the findings seriously hinders the scientific value. It is suggested that authors exteneively revise their manuscript to improve clarity and precision prior to consideration for publication.

Author Response

Responses to reviewer

Thanks to the reviewer for the useful comments that allow us to improve the quality of our paper by adding new data.

The reviewers’ comments are in italics. Sentences that are now included in the revised version of the paper are in inverted commas and highlighted in yellow in the manuscript. We have followed almost all the reviewers’ suggestions and modified the paper and references accordingly. 

The paper aims to address a significant issue, that is the carcinogenicity of childhood exposures to ambient particle-bound PAHs and heavy metals. It appears to be based on a human-subjects research (although not clear) that involves collection of biospecimens in the upper respiratory track from school-age children and measurements of size-fractionated particulate PAHs and heavy metals in the community. Unfortunately, the data are poorly analyzed. In addition, the study and data summaries are also poorly presented, diminishing the scientific value of the study. Specific examples include.

Some changes were added in Introduction section to better explain the aim and design of the study. See the answers below.

Similarly, we modified Materials and Methods and Results sections. The tables 1 and 2 of the old version were changed: the original data were maintained in the text and data partially already published in our previous papers were deleted. Data on metals in different sized fractions of particulate matter are reported in Table 3.

  1. The lack of description of the study design. A paper is cited [i.e. 26] however, a greater level of detail must be given for the readers to understand the study design and interpret the findings. It is not clear how samples (air and biospecimens) are collected, when, where, how many (schools and children in each school) and additional relevant information (such as co-exposures, diet, second hand exposure)

The Introduction section was modified to better explain that this is a conclusive paper of a complex and multidimensional evaluation of the relationship between air pollution parameters, in vitro mutagenicity tests and in vivo tests of early biological DNA damage in children.

We modified the lines 42-44 of the old version of Introduction section adding the following sentences:

“This study is the conclusion of the RESPIRA project that assessed the correlation between air pollution and early biological effects in exposed children. The chemical composition of PM and its genotoxicity, and the MN and comet tests on children’s buccal mucosa cells were reported in previous publications [16, 23, 26].

This paper aims to assess the relationships between the concentration and toxic equivalency of polycyclic aromatic hydrocarbons (PAHs), in different sized PM fractions collected near the schools attended by children, and air sample genotoxicity, and micronuclei and DNA damage (Comet test) in children’s buccal mucosa cells.

This study might add both theoretical and practical information on the effects of various air pollutants on children’s health and provide health-policies makers with a valuable tool for estimating the burden of disease attributable to urban air pollution. Because of the very large number of people exposed to air pollutants, even a small increase in the risk of disease is a relevant public health issue.”

Moreover, some details of the study design have been added in Materials and Methods section, highlighted in yellow.  

The RESPIRA study design involved recruiting 220 pre-school children, aged 3-6 years, in six schools located in different areas of town, to evaluate markers of early biological effect, such as primary DNA damage evaluated with the comet assay in salivary leukocytes and induction of micronuclei in exfoliated buccal cells, to investigate the genotoxic effects of air pollutants.

For the analysis carried out in this paper the data of only 132 subjects were used because those subjects had complete biological and questionnaire data. The subjects were divided in sampling cluster on the basis of biological sampling data and environmental sampling data (corresponding to 6 school sites).

The analysis on children’s biological markers was performed on the basis of attended school because the analysis on air pollutants was also carried out on school site. The children were distributed as follows: 18; 39; 32; 4; 20 and 19 from to school 1 to 6, respectively.

Actually, in our study we investigated other exposures, including indoor pollution, diet, physical activity, and others by questionnaires filled in by the children’s parents, which however showed no relationship with in vivo tests (Ceretti et al., 2014; Zani et al., 2020) and therefore, were not reported in this paper.

  1. The relevance of upper respiratory cells is not explained and the potential for contributions from other exposure pathways such as ingestion. It is known that due to their small size, particulate PAHs and heavy metals are less likely to be deposited in the upper respiratory track. On the other hand, cells in the oral region are regularly exposed to chemicals through digestion.

Buccal and nasal cells have been widely used in the biological monitoring of people exposed to airborne pollutants as they are representative of epithelial respiratory tract cells and are easier to collect than those of other respiratory organs. These cells can be used for assessing exposure to airborne mutagens, especially in the paediatric population as they are easy to collect. Many data are available on the comet and MN tests in peripheral blood leukocytes and several authors have found a positive association between DNA damage and exposure to air pollution. However, taking a blood sample in children to evaluate DNA damage can be considered as a very invasive procedure, possibly determining a low compliance of children and their parents. On the other hand, nasal or buccal cells may be a valid alternative. Given that the removal of the cells of the nasal mucosa could not be tolerated by some young children, because of their age, to encourage participation in the study we chose to collect buccal cells by brushing (to have epithelial cells for micronuclei test) and by spitting (to have leukocytes for comet assay).

  1. The description of the methods for the toxicological tests (replicates, sensitivity) and chemical analysis is missing. There are no information how PAHs were measured or heavy metals. This is particularly concerning, iven that PAHs concentrations are given in increments of 0.5 ng/m3 is an incosistent manner between Tables (table 2) and Figures.

The details of the in vitro and in vivo genotoxicity tests have been reported in previous papers.

The laboratory tests on particulate matter were performed in duplicate and all tests were repeated. The tests on biological samples were performed in duplicate.

Also the analytical procedures followed for PAHs and metals quantification have been reported in detail in a previous work. Therefore, we added the following statement in the revised version of Materials and Methods section: “Sampling and analytical procedures together with the obtained results are detailed in a previous paper, where the genotoxic activity of particulate matter and its possible correlations with PAHs and metals were assessed [16].”

We also added a new analysis on the basis of calculation of PAHs toxic equivalency for each PM fraction in different methods. In the revised version of Materials and Methods section we added the following sentences, accordingly the Results section was modified and tables 4 and 5 were added. 

“Regarding the PAHs, we calculated the concentration of total carcinogenic molecules as the sum of the compounds with definite or possible carcinogenic activity according to the IARC classification: benzo[a]anthracene (BaA), chrysene (CHR), benzo[b]fluoranthene (BbF), benzo[k]fluoranthene (BkF), benzo[j]fluoranthene (BjF), indeno[1,2,3-cd]pyrene (IcdP), dibenzo[a,h]anthracene (DahA), dibenzo[a,l]pyrene (DBalP), dibenzo[a,i]pyrene (DBaiP), dibenzo[a,h]pyrene (DBahP).

Furthermore, we computed the toxic equivalency (TEQ), as usually done for compounds with several congeners (for instance, dioxins), to assess the overall effect of the chemical mixture, using the most toxic one as the reference, with toxicity set to 1, based on epidemiological and toxicological literature data.

In particular, we calculated: 1) the total carcinogenic equivalency, CEQ (as proposed by Rogula-KozÅ‚owska et al. [27] after Nisbet and LaGoy [28], based on in vivo carcinogenicity data; 2) the toxic equivalency concentration, TEQ-DFG, according to the German Research Foundation (DFG), where a similar approach was followed, updated and improved including more congeners [29]; 3) the toxic equivalency of the mixtures,TCDD-TEQ, referred to the induction potency exerted by 2,3,7,8-tetrachlorodibenzo-p-dioxin (TCDD) [30]; 4) the BaP potency equivalence, BaP-PEQ, as indicated by the Minnesota Department of Health (MDH) [31]; 5) the mutagenic equivalence, MEQ, calculated, as reported by Rogula-KozÅ‚owska et al. [27]. Details of calculations are reported in the mentioned previous paper, where the toxic potentials of each sized fraction were compared [16].”

Therefore, five new references were added and consequently the numbering of the references has been changed.

  1. There are no indicators of repeated measurements (i.e. standard error) for both biological measurements and chemicals analysis. Is it because one sample was collected, or analyzed, no replicates.

The biological sampling was conducted only once. Also the environmental sampling by collecting particulate matter samples using high volume sampler was performed only once in each school area. The laboratory tests on particulate matter were performed in duplicate and all tests were repeated. The tests on biological samples were performed in duplicate.

The use of only one effect measure does not cause an important bias, however, because it reflects the mean exposure of the seven preceding days, when the climate situation had not substantially changed. On the other hand, previous studies which found an association between cytogenetic biomarkers and air pollution also used a single measure of effect.

Overall, the poor presentation and description of the findings seriously hinders the scientific value. It is suggested that authors extensively revise their manuscript to improve clarity and precision prior to consideration for publication.

See the answer to the previous comment 1. The Introduction section was modified to better explain that this is a conclusive paper, which evaluated the correlation between air pollution and early biological effects in children on already partially published data. For this reason, the detailed results of single tests were not reported in this paper because they were already published.

Reviewer 3 Report

Claudia Zani and colleagues provided an interesting work revealing the genotoxic activity of particulate matter and in vivo tests in chil-2 dren exposed to air pollution. It is a novel study, but some concern need to be addressed. 1. add more relative contents about necessity/research significance of present study in ‘introduction’ section 2. Iine49, ‘a sample of 132 pre-school children’ was stated. What’s the relationship of participant and 6 sites? What’ s the number of the sample of each site? Why just analysis the data of sites, not based on individual samples in next step? 3. add the R and p value of correlation analysis(figure1,figure2,figure3) 4. more detail of linear regression analysis should be provided(spearman? Periosn?), since the main methodology of this study was statistical analysis. 5.Usually, in environmental science study, concentrations of PAHs,metal,BaP was measured and repressed as means±SD/SEM. What’s the method and statistical analysis your group performing those measurements? 6. add profound discussion about the public health significance of present study.

Author Response

Thanks to the reviewer for the useful comments that allow us to improve the quality of our paper by adding new data.

The reviewers’ comments are in italics. Sentences that are now included in the revised version of the paper are in inverted commas and highlighted in yellow in the manuscript. We have followed almost all the reviewers’ suggestions and modified the paper and references accordingly. 

Claudia Zani and colleagues provided an interesting work revealing the genotoxic activity of particulate matter and in vivo tests in children exposed to air pollution. It is a novel study, but some concern need to be addressed.

  1. add more relative contents about necessity/research significance of present study in ‘introduction’ section

The Introduction section was modified to better explain that this is a conclusive paper on correlation between air pollution and early biological effects in children on already published data.

We modified the lines 42-44 of the old version of Introduction section adding the following sentences:

“This study is the conclusion of the RESPIRA project that assessed the correlation between air pollution and early biological effects in exposed children. The chemical composition of PM and its genotoxicity, and the MN and comet tests on children’s buccal mucosa cells were reported in previous publications [16, 23, 26].

This paper aims to assess the relationships between the concentration and toxic equivalency of polycyclic aromatic hydrocarbons (PAHs), in different sized PM fractions collected near the schools attended by children, and air sample genotoxicity, and micronuclei and DNA damage (Comet test) in children’s buccal mucosa cells.

This study might add both theoretical and practical information on the effects of various air pollutants on children’s health and provide health-policies makers with a valuable tool for estimating the burden of disease attributable to urban air pollution. Because of the very large number of people exposed to air pollutants, even a small increase in the risk of disease is a relevant public health issue”

Moreover, some details of the study design have been added in Materials and Methods section.  Actually, in our study we investigated other exposures, including indoor pollution, diet, physical activity, and others by questionnaires filled in by the children’s parents, which however showed no relationship with in vivo tests (Ceretti et al., 2014; Zani et al., 2020) and therefore, were not reported in this paper.

  1. line 49, ‘a sample of 132 pre-school children’ was stated. What’s the relationship of participant and 6 sites? What’s the number of the sample of each site? Why just analysis the data of sites, not based on individual samples in next step?

The RESPIRA study design involved recruiting 220 pre-school children, aged 3-6 years, in six schools located in different areas of town, to evaluate markers of early biological effect, such as primary DNA damage evaluated with the comet assay in salivary leukocytes and induction of micronuclei in exfoliated buccal cells, to investigate the genotoxic effects of air pollutants.

For the analysis carried out in this paper the data of only 132 subjects were used because those subjects had complete biological and questionnaire data. The subjects were divided in sampling cluster on the basis of biological sampling data and environmental sampling data (corresponding to 6 school sites).

The analysis on children’s biological markers was performed on the basis of attended school because the analysis on air pollutants was also carried out on school site. The children were distributed as follows: 18; 39; 32; 4; 20 and 19 from to school 1 to 6, respectively.

  1. add the R and p value of correlation analysis (figure1, figure2, figure3)

We modified Figures 1-3 adding the axis labels and measure units as suggested by reviewer.

The Spearman’s correlation coefficient rho and p values have been added to the Figures 1-3. The coefficients of linear regression ± standard error (SE) and p values were reported for each linear regression computed of MN in children mucosa buccal cells according to PAHs in the revised version in Table 5.

  1. more detail of linear regression analysis should be provided (spearman? Pearson?), since the main methodology of this study was statistical analysis.

In the revised version, we specified in Materials and Methods section that non-parametric correlation coefficients (Spearman rho) was used.

Lines 74-75 of the old version have been modified as follows: “Non-parametric correlation coefficient (Spearman rho) was used to assess possible correlation ….”.

Due to the non-normal distribution, we used the non-parametric Mann-Whitney test to evaluate the difference in micronuclei frequency between the children attending one school located next to a steel plant and those attending the other schools all together, due to consistently worse air quality parameters in the former than latter areas (Figure 4).

Lines 147-149 of the old version have been modified as follows: “Using the non-parametric Mann-Whitney test we found that the differences in micronuclei frequency between the children attending school 1 and those attending the other schools together were not statistically significant.”

The Figure 4 was simplified to focus on the differences in micronuclei frequency between the two groups of children.

  1. Usually, in environmental science study, concentrations of PAHs, metal, BaP was measured and repressed as means±SD/SEM. What’s the method and statistical analysis your group performing those measurements?

In the revised version the Table 2 has been modified by reporting only carcinogenic PAHs concentration. Concentration of total PAHs, BaP and metals measured at each sampling site in different fractions of PM was deleted. We added a new Table 3 with the mean of concentrations of metals measured in different fractions of PM.

We specified in Materials and Methods section that non-parametric correlation coefficient (Spearman rho) was used.

Also the analytical procedures followed for PAHs and metals quantification have been reported in detail in a previous work. Therefore, we added the following statement in new version of Materials and Methods section: “Sampling and analytical procedures together with the obtained results are detailed in a previous paper, where the genotoxic activity of particulate was assessed and possible correlations with PAHs and metals were searched [16].”

We also added a new analysis on the basis of calculation of PAHs toxic equivalency for each PM fraction in different methods. In the revised version of Materials and Methods we added the following sentences, accordingly the Results section was modified and tables 4 and 5 were added. 

“Regarding the PAHs, we calculated the concentration of total carcinogenic molecules as the sum of the compounds with definite or possible carcinogenic activity according to the IARC classification: benzo[a]anthracene (BaA), chrysene (CHR), benzo[b]fluoranthene (BbF), benzo[k]fluoranthene (BkF), benzo[j]fluoranthene (BjF), indeno[1,2,3-cd]pyrene (IcdP), dibenzo[a,h]anthracene (DahA), dibenzo[a,l]pyrene (DBalP), dibenzo[a,i]pyrene (DBaiP), dibenzo[a,h]pyrene (DBahP).

Furthermore, we computed the toxic equivalency (TEQ), as usually done for compounds with several congeners (for instance, dioxins), to assess the overall effect of the chemical mixture, using the most toxic one as the reference, with toxicity set to 1, based on epidemiological and toxicological literature data.

In particular, we calculated: 1) the total carcinogenic equivalency, CEQ (as proposed by Rogula-KozÅ‚owska et al. [27] after Nisbet and LaGoy [28], based on in vivo carcinogenicity data; 2) the toxic equivalency concentration, TEQ-DFG, according to the German Research Foundation (DFG), where a similar approach was followed, updated and improved including more congeners [29]; 3) the toxic equivalency of the mixtures,TCDD-TEQ, referred to the induction potency exerted by 2,3,7,8-tetrachlorodibenzo-p-dioxin (TCDD) [30]; 4) the BaP potency equivalence, BaP-PEQ, as indicated by the Minnesota Department of Health (MDH) [31]; 5) the mutagenic equivalence, MEQ, calculated, as reported by Rogula-KozÅ‚owska et al. [27]. Details of calculations are reported in the mentioned previous paper, where the toxic potentials of each sized fraction were compared [16].”

Therefore, five new references were added and consequently the numbering of the references has been changed.

  1. add profound discussion about the public health significance of present study.

The public health significance of this study was more emphasized in the Introduction and Discussion sections. Regarding the Introduction section see the response to the previous comment. Moreover, we added in Discussion section the following sentences:

“In our study, we found that the in vivo tests for detecting early biological effect in children were associated with the concentration of some air pollutants, and therefore they could be used as simple, rapid and low-cost tests for evaluating specific environmental situations and also the impact of possible interventions for contrasting the health effects of air pollution. In particular MN test, being predictive of the presence of health effects in children, could be proposed in addition to the traditional parameters for the monitoring of urban air pollution.”

Round 2

Reviewer 1 Report

The authors have addressed all comments and improved the manuscript.

I think, the manuscript can be published in the  present form.

However,  the novelty of the work is still not clearly specified.

Author Response

Responses to reviewers

Thanks to the reviewers for their comments. We are pleased to have fulfilled their requests.

The reviewers’ comments are in italics. Sentences that are now included in the revised version of the paper are in inverted commas and highlighted in yellow in the manuscript.

Comments and Suggestions for Authors

The authors have addressed all comments and improved the manuscript.

I think, the manuscript can be published in the present form.

However, the novelty of the work is still not clearly specified.

Thanks to the reviewer for his comment. We are pleased to have fulfilled his requests.

We added the following sentence in the Introduction section (lines 50-52 of the revised version):

“The main novelty of this research is the use of these measures of early biological effect in a considerable number of young children living in a heavy polluted area, which can be detected a long time before clinical disease develops.”

Reviewer 2 Report

The authors addressed my previous concerns

Author Response

Responses to reviewers

Thanks to the reviewers for their comments. The reviewers’ comments are in italics. Sentences that are now included in the revised version of the paper are in inverted commas and highlighted in yellow in the manuscript.

Comments and Suggestions for Authors

The authors addressed my previous concerns.

Thanks to the reviewer for his comment. We are pleased to have fulfilled his requests.

Reviewer 3 Report

All the comments were well adressed .The MS currently meets the requirements of publishing 

Author Response

Responses to reviewers

Thanks to the reviewers for their comments. The reviewers’ comments are in italics. Sentences that are now included in the revised version of the paper are in inverted commas and highlighted in yellow in the manuscript.

Comments and Suggestions for Authors

All the comments were well adressed. The MS currently meets the requirements of publishing.

Thanks to the reviewer for his comment. We are pleased to have fulfilled his requests.